# High-throughput proteomics of nanogram-scale samples with Zeno SWATH MS

**Ziyue Wang[1], Michael Mülleder[2], Ihor Batruch[3], Anjali Chelur[3], Kathrin Textoris-Taube[1,2], Torsten Schwecke[1], Johannes Hartl[1], Jason Causon[3], Jose Castro-Perez[3], Vadim Demichev[1], Stephen Tate[3], Markus Ralser[1,4]\***

[1]Department of Biochemistry, Charité – Universitätsmedizin Berlin, Corporate Member of Freie Universität Berlin and Humboldt-Universität zu Berlin, Berlin, Germany; [2]Core Facility – High-Throughput Mass Spectrometry, Charité – Universitätsmedizin Berlin, Corporate Member of Freie Universität Berlin and Humboldt-Universität zu Berlin, Core Facility – High-Throughput Mass Spectrometry, Berlin, Germany; [3]Sciex, Concord, Canada; [4]The Wellcome Centre for Human Genetics, Nuffield Department of Medicine, University of Oxford, Oxford, United Kingdom

**Abstract** The possibility to record proteomes in high throughput and at high quality has opened new avenues for biomedical research, drug discovery, systems biology, and clinical translation. However, high-throughput proteomic experiments often require high sample amounts and can be less sensitive compared to conventional proteomic experiments. Here, we introduce and benchmark Zeno SWATH MS, a data-independent acquisition technique that employs a linear ion trap pulsing (Zeno trap pulsing) to increase the sensitivity in high-throughput proteomic experiments. We demonstrate that when combined with fast micro- or analytical flow-rate chromatography, Zeno SWATH MS increases protein identification with low sample amounts. For instance, using 20 min micro-flow-rate chromatography, Zeno SWATH MS identified more than 5000 proteins consistently, and with a coefficient of variation of 6%, from a 62.5 ng load of human cell line tryptic digest. Using 5 min analytical flow-rate chromatography (800 µl/min), Zeno SWATH MS identified 4907 proteins from a triplicate injection of 2 µg of a human cell lysate, or more than 3000 proteins from a 250 ng tryptic digest. Zeno SWATH MS hence facilitates sensitive high-throughput proteomic experiments with low sample amounts, mitigating the current bottlenecks of high-throughput proteomics.

\*For correspondence:
markus.ralser@charite.de

## Editor's evaluation

This paper presents a novel method for high-throughput proteomic data. This is an important improvement allowing a 5-8 fold increase in detection sensitivity over methods of similar throughput. A compelling set of data acquired on human and yeast whole-cell lysate illustrate this gain in performance.

## Introduction

The proteome consists of large numbers of proteins that differ in concentration, distribution, and temporal dynamics (*Eriksson and Fenyö, 2007*; *Hixson et al., 2017*). The study of the proteome hence requires the ability to address both its large dynamic range and complexity, as well as the dynamic nature of biological systems. While proteomics typically aims to quantify as many proteins as

possible, many of the biological questions that are to be addressed also necessitate the processing of large sample series. For example, a minimal drug treatment experiment of a wild-type and a single mutant cell line, with five drug concentrations, five time points, and triplicate measurements, already produces 150 samples. Epidemiological studies, exemplified by the Fenland study (*O'Connor et al., 2015*), can reach tens of thousands of samples. Processing such high sample amounts requires fast and robust measurement technologies that are quantification precise and cost effective.

In bottom-up proteomics, data-dependent acquisition (DDA) selects precursors from full-scan MS[1] mass spectrum for fragmentation, generating tandem (MS/MS or MS[2]) mass spectra that can be matched to spectra in a database. While DDA ensures selectivity, it can result in high numbers of missing values that can originate from, for example, the stochastic nature of precursors selected for fragmentation, which limits the application in high-throughput studies. Data-independent acquisition (DIA) techniques such as SWATH acquisition (*Gillet et al., 2012*) address this problem by generating fragment ion spectra from all precursor ions that fall within a predefined precursor m/z range window. DIA shifted the data-processing strategies from an established spectrum-centric processing model to a peptide-centric model, requiring a shift in algorithm design and assumptions. DIA methods have facilitated applications in large-scale proteomics, including systems biology studies in various model organisms, disease states, and species (*Hou et al., 2015*; *Gao et al., 2017*; *Basak et al., 2015*; *Zelezniak et al., 2018*; *Grobbler et al., 2015*; *Gillet et al., 2012*), with one of the most common clinical applications being the discovery of (compound) biomarkers for disease in body fluids, particularly blood serum or plasma (*Anjo et al., 2017*; *Gajbhiye et al., 2016*; *Russell et al., 2016*; *Hou et al., 2015*; *Pires et al., 2016*).

Throughput, the required measurement precision, and sample amount are major factors that need to be balanced when conducting large-scale proteomics measurements. Moreover, particular attention has to be given to batch effects that exist within large proteomic experiments. Proteomics has historically relied on nano-flow chromatography, which is highly sensitive as sample dilution is minimal (*Muntel et al., 2019*), but the low flow rate requires a longer gradient time, which is more susceptible to technical distortions. Moreover, columns compatible with nano-LC systems typically require exchange after a maximum of a few hundred injections, which together prolongs the acquisition time and leads to complex batch effects in larger studies. Recent advances in instrumentation and software have facilitated the use of faster gradients and higher chromatographic flow rates (*Messner et al., 2020*; *Bian et al., 2020*; *Vowinckel et al., 2018*). These accelerate proteomic experiments and allow highly precise measurements. These come, however, at the cost of a higher sample amount required and can yield lower proteomic depth.

Here, we present a new acquisition technique, Zeno SWATH MS, that exploits a linear ion trap, the Zeno trap (*Loboda and Chernushevich, 2009*), in combination with SWATH MS acquisition, in order to increase the sensitivity of DIA-MS acquisition, specifically in combination with micro-flow-rate and analytical flow-rate chromatography. We demonstrate that Zeno SWATH MS increases sensitivity and expands the proteomic dynamic range coverage, generating new possibilities in the design of high-throughput proteomic experiments.

## Results

### Zeno SWATH MS increases proteomic depth and quantification precision

The sensitivity of SWATH MS in time-of-flight (TOF) mass spectrometers is largely determined by the MS/MS duty cycle. The pulse in the TOF accelerator, which results in a small proportion of ions situated at the right position of entering TOF, has a disproportionate impact on lower-m/z ions, resulting in a duty cycle of TOFs of only 5–25%. Here, we describe a setup of the Zeno trap (*Loboda and Chernushevich, 2009*; *Figure 1—figure supplement 1*), where all fragment ions are trapped in an axial pseudopotential well and then released by potential energy with timing aligned to the next pulse at the accelerator, allowing a duty cycle increase, thus increasing intensity 4–20 times (*Baba et al., 2021*). We speculated that the Zeno trap could be employed in combination with SWATH MS to boost the MS/MS sensitivity, facilitating higher identification and quantification performance while requiring lower sample amounts.

To generate platforms for medium- and high-throughput proteomic experiments, we coupled a ZenoTOF 7600 system (SCIEX) equipped with a Zeno trap (*Figure 1—figure supplement 1*) to both a micro-flow LC system (ACQUITY M-Class, Waters) and an analytical flow-rate LC (1290 Infinity II, Agilent), respectively. For data processing, we applied our deep-neural-network-based software suite, DIA-NN, which has been specifically optimised for handling complex data arising from fast chromatographic methods (*Demichev et al., 2020*).

First, we examined how consistently precursor (i.e. peptide at a specific charge state) and protein intensities were measured between triplicate injections of 62.5 ng K562 human cell line standard, comparing the performance of Zeno SWATH MS with SWATH MS on the same instrument setup. Peptides were separated using the micro-flow LC running at a flow rate of 5 µl/min, 20 min micro-flow gradient chromatography (Materials and methods). For analysis, we used an experimental spectral library generated using the ZenoTOF 7600 system (Materials and methods). Zeno SWATH MS yielded 5179 consistently quantified proteins among an average of 6226 proteins identified in triplicates. These were quantified with a median coefficient of variance (CV) of 6%, and 89% of identifications or 4602 shared proteins were quantified with a CV better than 20% (*Figure 1a*). At the precursor level, on average 49,172 precursors were identified, of which 40,209 were repeatedly quantified. Of these, 77% were quantified with a CV of less than 20% (*Figure 1b*). Zeno SWATH MS showed a considerable improvement in quantification consistency compared to conventional SWATH MS using the same chromatography, mass spectrometry, and sample. SWATH MS consistently quantified 2743 proteins, of which 76% or 2083 proteins were quantified with a CV <20% (*Figure 1a*). At the precursor level, SWATH identified an average of 21,771 precursors in triplicate injections, of which 16,022 were consistently quantified, with a median CV of 11%.

The analysis of proteomes of different species mixed in different ratios, as introduced with the LFQbench test (*Navarro et al., 2016*; *Kuharev et al., 2015*), has proven an efficient way for evaluating the capabilities of DIA workflows and data-processing software (*Demichev et al., 2020*; *Bruderer et al., 2015*). The multiple-species benchmarks evaluate both the identification and quantification precision, which is determined by the ability of the proteomic method to deconvolute the multiplexed proteomes. Herein, we assessed how well the known ratios between two-species lysates – human and yeast lysates mixed in the defined proportions: 1:1 (A/B) for human, 2:1 for yeast (Materials and methods) – were recovered in triplicates using SWATH MS and Zeno SWATH MS, followed by a library-free analysis in DIA-NN. We injected triplicates of 1 µl of both mixtures (65 ng [mixture A] and 47.5 ng [mixture B]) with micro-flow-rate chromatography. Compared with SWATH MS, Zeno SWATH MS revealed more precursors and proteins quantified at better quantification precision. At the precursor level, Zeno SWATH MS quantified 34,872 human and 15,610 yeast precursors, while SWATH quantified 16,167 human and 6847 yeast precursors (SWATH MS, *Figure 1—figure supplement 2b*; Zeno SWATH MS, *Figure 1—figure supplement 2c*). At the protein level, 3414 (human) and 1816 (yeast) proteins were quantified with Zeno SWATH MS, while 2064 human and 973 yeast proteins were quantified using SWATH MS. Further, 3282 human and 1692 yeast proteins were quantified with Zeno SWATH MS, showing a ratio close to the known ratio (1:1 (A/B) for human, 2:1 for yeast) in at least two replicates in each of mixture A and B, with a median CV value for human proteins measured in triplicates of mixture A and B of 6.9% and 7%, respectively. The human proteins measured with SWATH MS show a median CV value of 12.7% and 12.6% for triplicate mixture A and B, with 1741 human and 776 yeast proteins quantification close to the known ratio (Zeno SWATH MS, *Figure 1c*; SWATH, *Figure 1—figure supplement 2a*). To exclude the bias introduced by data normalisation, we also compared the non-normalised data in both acquisition methods (SWATH MS, *Figure 1—figure supplement 2d*; Zeno SWATH MS, *Figure 1—figure supplement 2e*).

In order to account for the protein amount's effect on performance of the two-species benchmarks, we conducted a similar analysis with a 50% lower injection amount. Compared with SWATH MS, 118% more human and 133% more yeast proteins were quantified in Zeno SWATH MS with 0.5 µl injection; while 65% more human and 86% more yeast proteins quantified in Zeno SWATH MS were quantified with 1 µl injection (SWATH MS, *Figure 1—figure supplement 2f*; Zeno SWATH MS, *Figure 1—figure supplement 2g*). Moreover, when comparing the ratio of human and yeast proteome quantified in both mixtures ($log_2$(A/B)), Zeno SWATH MS shows less inter-species interference and is able to represent the real quantification ratio in both species more accurately ($log_2$(A/B)=0 (human), = 1 (yeast)) (*Figure 1d*).

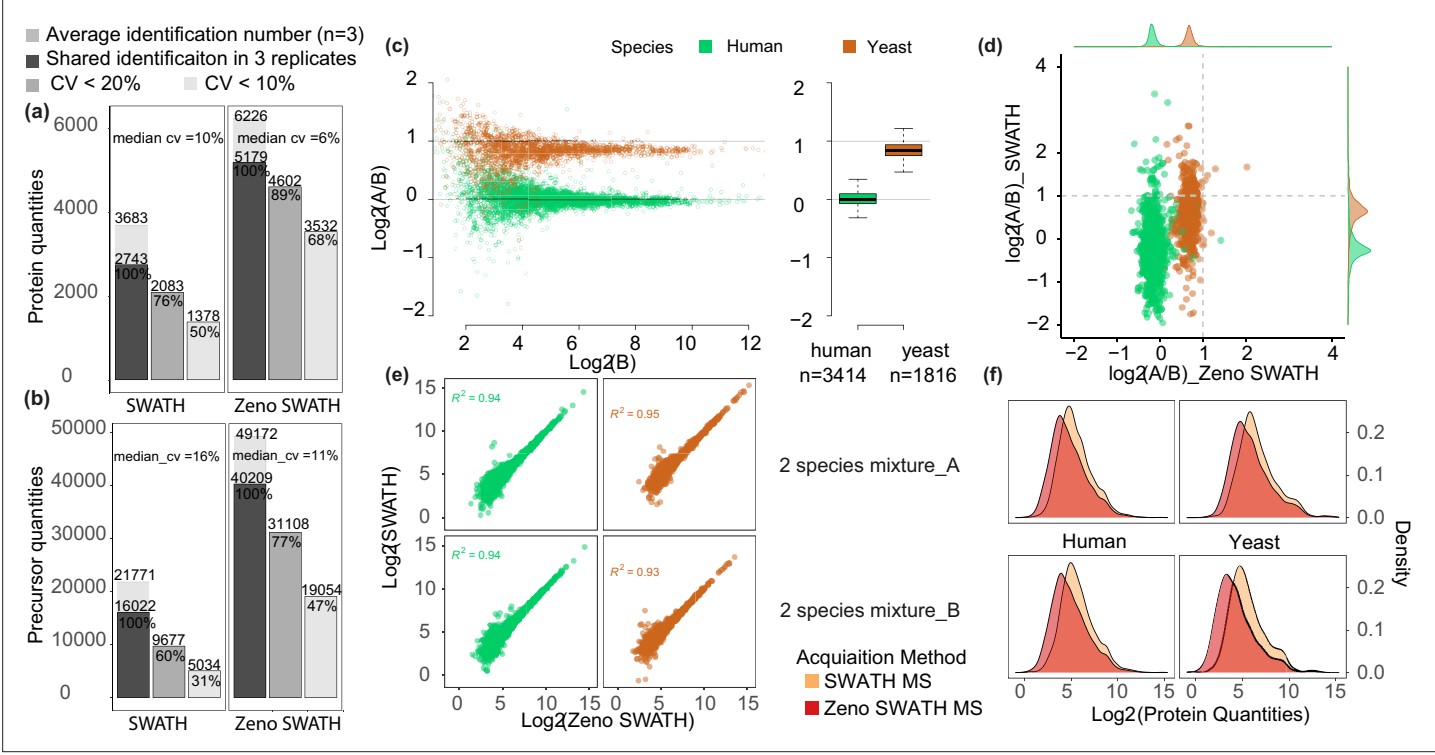

**Figure 1.** Comparison between SWATH and Zeno SWATH MS in proteome coverage and quantification precision and using 5 µl/min, 20 min micro-flow-rate chromatography. (**a,b**) Reproducibility of protein identification using SWATH MS and Zeno SWATH MS on human cell line standard (K562) separated by micro-flow chromatography with a 62.5 ng load. Average identification numbers of proteins (**a**) and precursors (**b**) across three technical replicates (grey background bar) from a human cell line (K562) proteome tryptic digest are given; numbers of consistent identifications in technical replicates are given in dark grey; proteins or precursors quantified with coefficient of variation (CV) better than 20% in grey, and those quantified with a CV better than 10% in light grey. (**c**) Protein-level LFQbench results for Zeno SWATH MS. Quantification precision was benchmarked using yeast proteome tryptic digests that were spiked in two different proportions (A and B, three repeat injections each) into human cell line standard (K562) (A: 30 ng K562 + 35 ng yeast; B: 30 ng K562 + 17.5 ng yeast). Raw data were processed by library-free mode DIA-NN analysis. Protein ratios between the mixtures were visualised using the LFQbench R package (**Navarro et al., 2016**). Black dashed lines represent the expected $\log_2(A/B)$ values for human (=0) and yeast (=1). Left pane, log-transformed ratios ($\log_2(A/B)$) of proteins plotted for DIA-NN over the log-transformed intensity of sample B. Right pane, protein quantification performance shown as box plots (boxes, interquartile range; whiskers, 1–99 percentile; n=3414 [human] and n=1816 [yeast]). (**d**) Average protein quantity differences between mixture A and B. Compared to SWATH MS, Zeno SWATH MS shows an improved separation of the two-species proteomes and more accurately represents the real ratio of the precursors ($\log_2$ of human quantity ratio close to 0, while $\log_2$ of yeast quantity ratio close to 1). (**e**) Correlation of protein quantities between both MS schemes in different mixtures. Human and yeast protein quantities in both mixtures show a high correlation between methods. (**f**) Distribution of protein quantities of each species in each mixture (A and B). In both mixtures and both species, Zeno SWATH MS shows deeper proteomic coverage by quantifying more low-abundant proteins.

The online version of this article includes the following figure supplement(s) for figure 1:

**Figure supplement 1.** The ZenoTOF structure and Zeno Trap used in Zeno SWATH.

**Figure supplement 2.** Comparison between SWATH MS and Zeno SWATH MS in quantification precision with two-species proteome mixture analysis.

We speculated that the more accurate quantification ratio provided by Zeno SWATH MS comes from the better identification and more accurate quantification of the lower-abundant proteins. We therefore correlated the human and yeast proteomes shared in both acquisition methods (scatter plot, *Figure 1e*). For both species, the quantification results from SWATH MS and Zeno SWATH MS are in very good agreement for high-abundant proteins, but diverge for low-abundant ones. Then, we compared the protein quantities of SWATH MS and Zeno SWATH MS in both species (density plot, *Figure 1f*). In both species, Zeno SWATH MS performs better in quantifying the lower-abundant proteins.

## Zeno SWATH MS increases protein identification rate with low sample amounts

Next, we tested the sensitivity of Zeno SWATH MS by analysing a concentration series of the K562 human cell line standardusing both SWATH MS and Zeno SWATH MS. The concentration series was

acquired starting with the lowest concentrations injected after several blank measurements with a 20 min, 5 µl/min chromatographic gradient (Materials and methods), covering sample amounts ranging from near 1 to 500 ng (Materials and methods). For a comparison of MS systems, a similar concentration series was acquired using SWATH MS on a TripleTOF 6600 system (SCIEX) with the same 5 µl/min, 20 min chromatography setup (Materials and methods). To avoid the potential bias that system-specific experimental spectral libraries have on data from different MS systems, the benchmark was carried out using library-free data analysis.

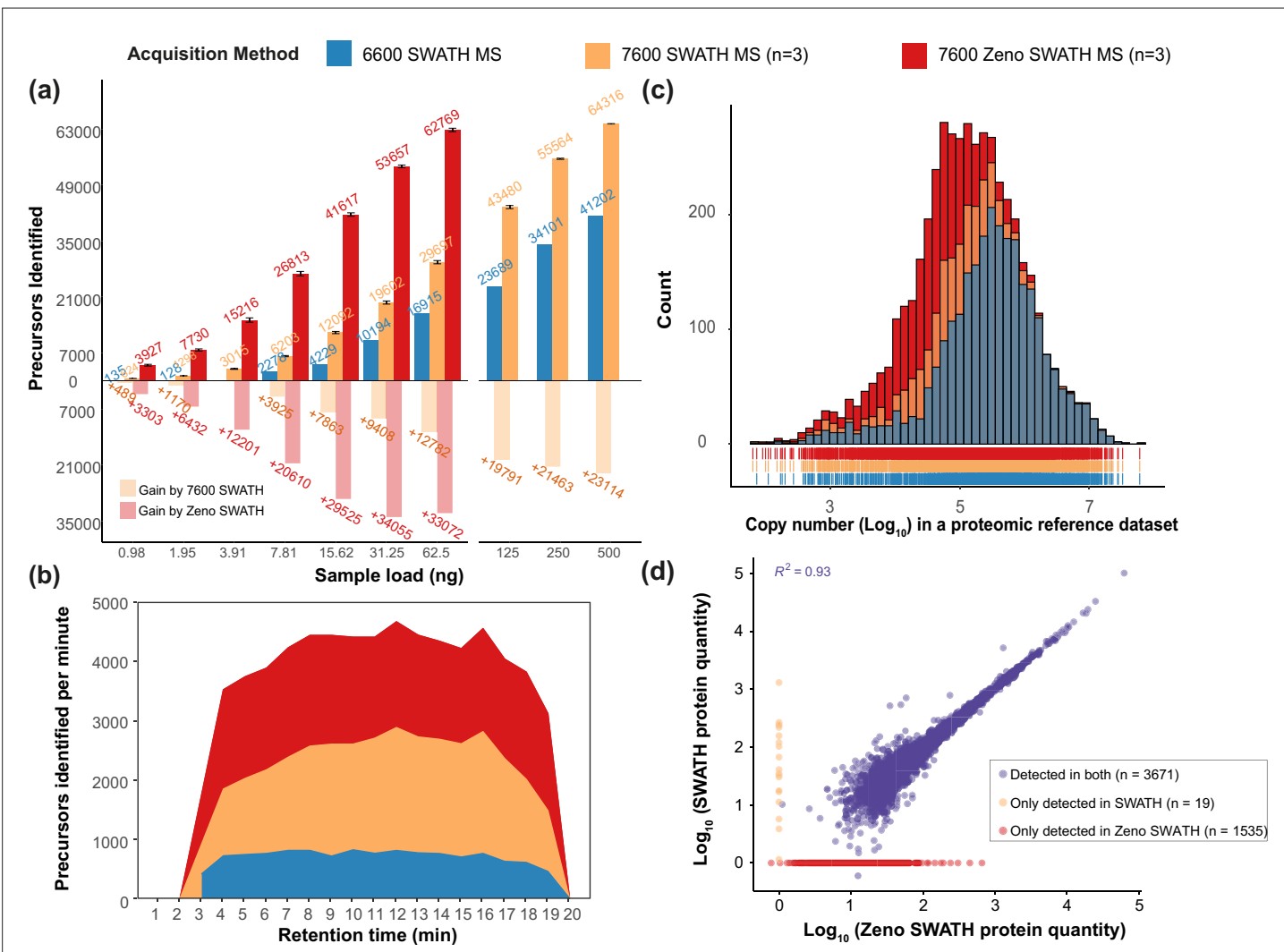

**Figure 2.** Zeno SWATH MS and its performance on K562 human cell line standard using 5 µl/min, 20 min micro-flow chromatography (SWATH MS on the TripleTOF 6600 system [blue], SWATH MS on the ZenoTOF 7600 system [yellow], and Zeno SWATH MS on the ZenoTOF 7600 system [red]). (**a**) Precursor identification performance using SWATH and Zeno SWATH. Illustrated is the average number of precursors identified (n=3, error bars [±1 SD] shown in plot) for a K562 standard dilution series under three acquisition methods with library-free DIA-NN analysis (second-pass data). (**b**) Visualisation of precursor identification across gradient time with SWATH and Zeno SWATH MS of 62.5 ng K562 standard injection. (**c**) Histogram (bins = 50) of the protein abundance distribution represented as copy number in $\log_{10}$ scale of a human proteomic reference dataset (***Bekker-Jensen et al., 2017***). Zeno SWATH MS increases protein identification numbers by quantifying more low-abundant proteins. (**d**) Correlation of protein quantity between SWATH and Zeno SWATH MS. Illustrated is the correlation of protein quantity between two MS acquisition schemes; 3671 proteins. proteins were identified and quantified in both acquisition schemes, 19 proteins only quantified with SWATH MS, 1535 proteins quantified only by Zeno SWATH MS. Among the 3671 proteins quantified by both acquisition schemes, there is a high correlation of protein quantities between the proteins identified in both acquisition schemes ($r^2$=0.93.).

The online version of this article includes the following figure supplement(s) for figure 2:

**Figure supplement 1.** Zeno SWATH MS and its performance on the K562 human cell line standard using 5 µl/min, 20 min micro-flow chromatography (SWATH MS on the TripleTOF 6600 system [blue], SWATH MS on the ZenoTOF 7600 system [yellow], and Zeno SWATH MS [red]).

Zeno SWATH MS consistently yielded better quantification performance. For instance, from the 62.5 ng K562 human cell line standard, 16,915 precursors or 2886 proteins were identified using the TripleTOF 6600 system at a 1% false discovery rate (*Figure 2a*). Using SWATH MS on the 7600 instrument, these numbers increased to 29,697 precursors identified from three technical replicates and an average of 3724 proteins (*Figure 2a*; protein identification in *Figure 2—figure supplement 1a*, first-pass precursor identification in *Figure 2—figure supplement 1b*). With Zeno SWATH MS on the same 7600 setup, an average of 62,769 precursors, or 5206 proteins, were identified from triplicate injections of 62.5 ng of the K562 human cell line standard. Interpreting the data in a different way, Zeno SWATH MS identified a similar number of precursors and proteins from 62.5 ng of K562 human cell line standard as SWATH MS did from 500 ng of the same sample on the same instrument (SWATH identified 64,316 precursors or 5242 proteins from 500 ng) (*Figure 2a*). Normalised per gradient time, this means that up to 4636 precursors were detected per minute of active runtime in Zeno SWATH MS, whereas SWATH from the ZenoTOF 7600 system quantified 2874 precursors/min and SWATH from the TripleTOF 6600 system reached 1153 precursors/min (*Figure 2b*).

We speculated that the identification gains could be contributed by the better identification of low-abundant proteins by Zeno SWATH MS. We therefore assessed the proteome concentration dynamic range of the proteins covered by comparing our K562 human cell line standard results to a proteomic reference dataset with copy-number estimates for 14,178 human protein groups (*Bekker-Jensen et al., 2017*). SWATH MS on the 7600 system (3690 protein groups quantified) showed a larger protein concentration dynamic range coverage compared to the TripleTOF 6600 system (2881 protein groups quantified). Zeno SWATH MS increased the number of proteins identified (5206 protein groups quantified). Comparison with the standard dataset revealed that the gains are due to more low-abundant proteins quantified (*Figure 2c*). Also here, we report a high agreement of protein quantities between methods overall ($r^2$=0.93), and in particular among high-abundant proteins (*Figure 2d*).

## Zeno SWATH MS facilitates high-throughput analytical flow-rate proteomics with low sample amounts

Because of its high analytical performance and stability in combination with short chromatographic gradients, analytical flow-rate chromatography is a method of choice for large-scale projects and in diagnostic laboratories. Analytical flow-rate chromatography is also attractive for practical reasons, because columns equilibrate quickly, and the problem of dead volumes and sample carryover is substantially reduced. In our recent studies, we reduced the time between injections to 3 min or less (*Messner et al., 2021*). Specifically, for applications such as neat-plasma proteomics or drug screens, we have previously demonstrated proteomic experiments using flow rates of 800 μl/min, and obtained quantification of precise proteomes with gradient times that ranged from 30 s to 5 min (*Messner et al., 2020*; *Messner et al., 2021*). Consequently, even with the slowest of these gradients – 5 min of active separation – the throughput of a proteomics study is increased to 180 samples per day on a single instrument. The downside so far is a reliance on relatively large sample amounts, which restricts the application space and shortens the instrument cleaning cycles. We therefore examined whether Zeno SWATH MS's performance would allow us to conduct proteomic experiments with analytical flow-rate chromatography and lower sample amounts.

We used the ZenoTOF system coupled to the 1290 Infinity II HPLC system (Agilent), running at 800 μl/min, with a 5 min gradient chromatography. We analysed the K562 human cell line standard in a concentration series, injecting increasing amounts from 4 ng to 2 μg. The obtained raw data were analysed using a spectral library. On average, Zeno SWATH MS identified 29,681 precursors and 4907 proteins with a 2 μg load of sample on the column, whereas SWATH MS identified 17,212 precursors or 3250 proteins with the same amount of sample (protein data shown in *Figure 3a*, respective precursor data in *Figure 3—figure supplement 1a*). Importantly, with Zeno SWATH MS, proteomic depth was reached with analytical flow-rate chromatography even with low sample amounts. For instance, with 250 ng of K562 human cell line standard injected, around 3000 proteins were quantified with Zeno SWATH MS while a similar protein quantification number can only be reached by SWATH MS with 2 μg injection on the same instrument and chromatography (*Figure 3a*). Further, the stability of the analytical flow-rate LC system was reflected in the consistency of the quantification and yielded a median protein quantification CV of 6%. Expressed differently, 92% of the proteins were consistently identified with a CV <20% (*Figure 3b*, respective precursor data in *Figure 3—figure supplement 1b*).

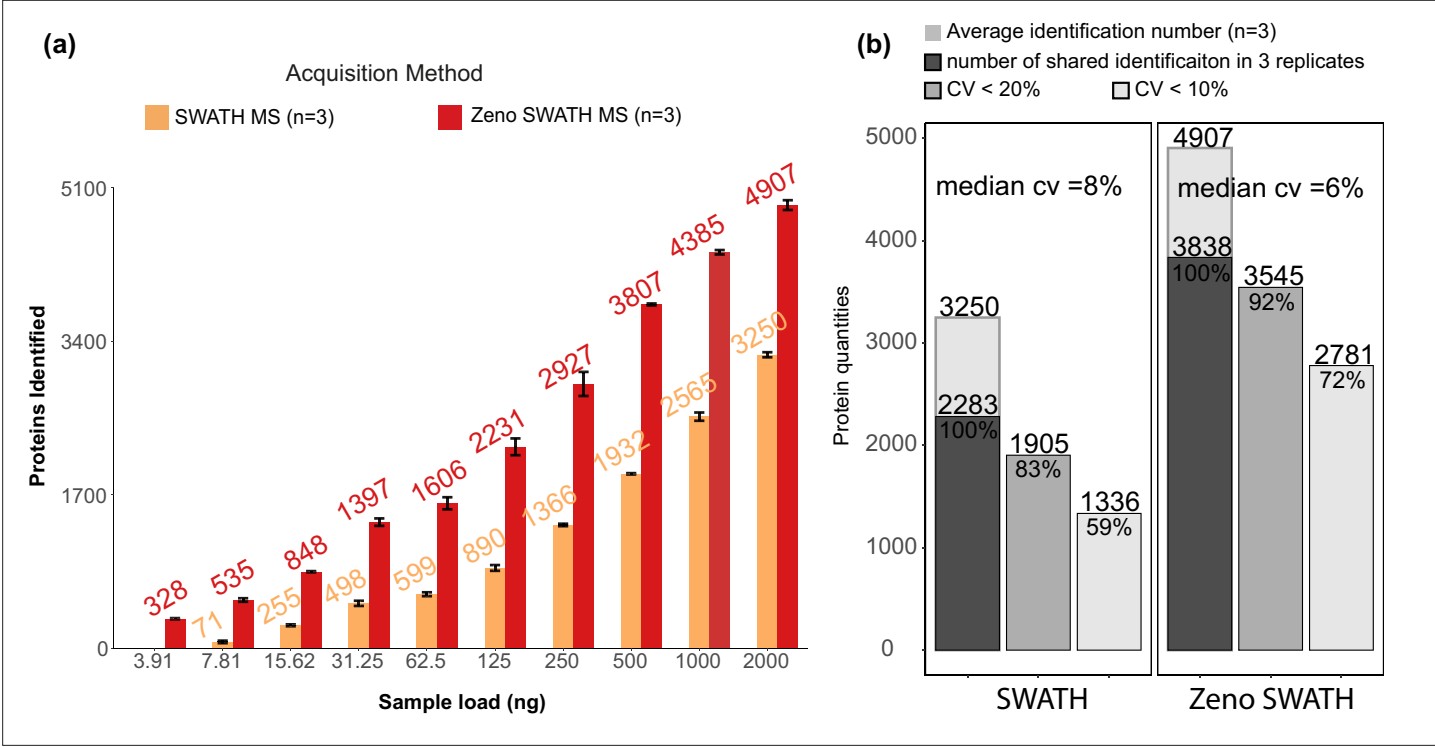

**Figure 3.** Protein identification performance on K562 human cell line standard using 800 μl/min, 5 min gradient chromatography in SWATH and Zeno SWATH MS. (**a**) Comparison of sample injection amount and identification performance with SWATH and Zeno SWATH MS. Illustrated is the average number of proteins identified (n=3, error bars [±1 SD] shown in plot) for a K562 human cell line standard dilution series under two acquisition methods with spectral library DIA-NN analysis. (**b**) Reproducibility of proteins quantified in SWATH and Zeno SWATH MS on 2 μg of K562 human cell line standard separated by analytical flow-rate chromatography. Average protein identification number in three replicates (background), number of proteins identified in all replicates (dark grey), proteins with coefficient of variation below 20% (grey) and below 10% (light grey) of SWATH and Zeno SWATH MS. Raw data were analysed by spectral library-based DIA-NN analysis.

The online version of this article includes the following figure supplement(s) for figure 3:

**Figure supplement 1.** Precursor identification performance on K562 human cell line standard using 800 μl/min, 5 min gradient chromatography in SWATH and Zeno SWATH MS.

## Evaluation of Zeno SWATH MS on plasma, plant, fungal, and bacterial samples

A key application space for high-throughput proteomics is human plasma and serum proteomics, as well as systems and synthetic biology. Plasma, bacterial, plant, and fungal sample matrices differ, however, in dynamic range and proteomic complexity from a human cell line extract such as the K562 human cell line standard. For instance, in human plasma, 99.9% of the proteomic mass is attributable to less than 200 proteins (*Anderson and Anderson, 2002*), while a yeast proteome coverage saturates at around 4000 proteins even with prefractionation (*Thakur et al., 2011*). To assess the performance of Zeno SWATH MS on various matrices, we hence assayed a concentration series (1–500 ng) of human plasma as well as cell extracts from the yeast *Saccharomyces cerevisiae*, the bacterium *Escherichia coli,* and seedlings of chickpea (*Cicer arietinum*). The standards and samples generated were separated using 20 min micro-flow-rate chromatography (Materials and methods) and acquired with both Zeno SWATH MS and SWATH MS.

In non-depleted human plasma, Zeno SWATH MS identified 3957 precursors derived from 204 proteins with a 62.5 ng injection. In human neat plasma, Zeno SWATH MS identified 3947 precursors from 245 proteins, while with SWATH MS, 2516 precursors from 156 proteins were identified from the same amount of sample. Notably, 15 ng of sample was sufficient to quantify the dominating fraction of the plasma proteome (160 proteins) (*Figure 4*). Zeno SWATH MS could hence be applied to high-throughput plasma samples where sample amounts are limited, for instance to make proteomics applicable for analysing finger-prick samples or in combination with nano-sampling devices.

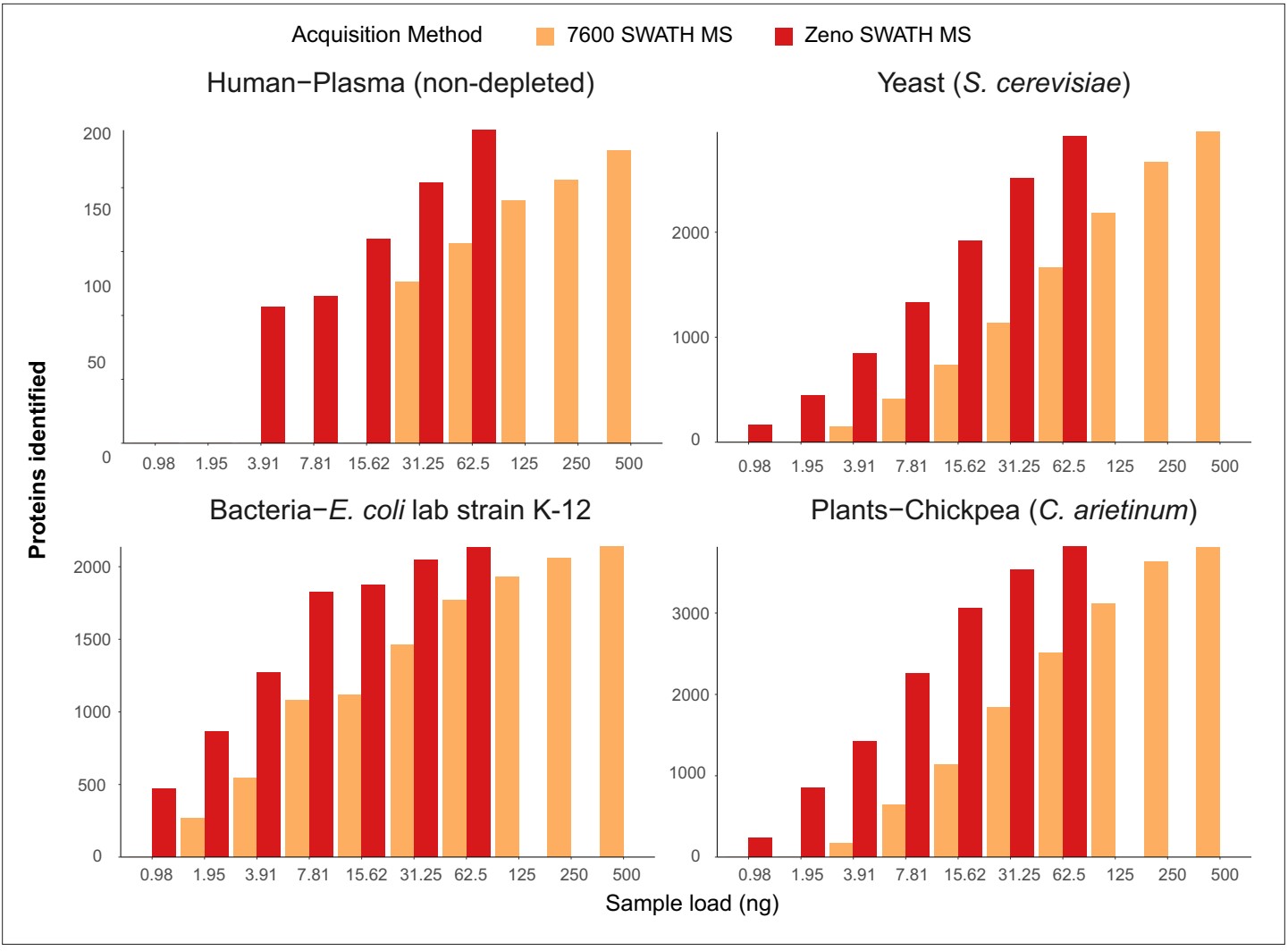

**Figure 4.** Protein identification number with SWATH MS and Zeno SWATH MS in different sample types with 5 µl/min, 20 min micro-flow-rate chromatography. We generated tryptic digests from human plasma, and protein extracts from the yeast *Saccharomyces cerevisiae* S228c, the *Escherichia coli* lab strain K-12, and chickpea (*Cicer arietinum*) seedlings germinated in the lab. Data were processed with library-free DIA-NN analysis. Shown data are the identification numbers of and concentration, detailed information of numbers can be found in ***Supplementary file 1***.

The online version of this article includes the following figure supplement(s) for figure 4:

**Figure supplement 1.** Precursor identification with SWATH acquisition and Zeno SWATH MS in different sample types with 5 µl/min, 20 min micro-flow-rate chromatography.

*S. cerevisiae* is one of the dominating workhorses of biotechnology, a key model organism for basic research, closely related to important clinical pathogens such as *Candida glabrata* (***Oliver, 2022***), and a pathogen on its own right. In a *S. cerevisiae* protein extract tryptic digest, micro-flow-rate Zeno SWATH MS quantified 29,339 precursors from 2915 proteins from a 62.5 ng injection, compared to 1660 proteins quantified by SWATH on the same instrument setup. From an extract of *E. coli*, the dominating bacterial model organism, Zeno SWATH MS identified 24,880 precursors or 2131 proteins at a 62.5 ng load, whereas conventional SWATH acquired 16,139 and 1768 precursors and proteins, respectively. And finally, we identified 33,694 precursors and 3815 proteins in 62.5 ng of chickpea (*C. arietinum*) seedling extract, chosen as a representative sample for a plant proteome. In comparison, 500 ng of sample was needed to reach similar identification numbers (34,573 precursors, 3807 proteins) using SWATH acquisition on the plant proteome tryptic digest (an overview of achieved protein identification is shown in ***Figure 4***; details for respective precursor identification are shown in

*Figure 4—figure supplement 1*; identification numbers for both proteins and precursors are shown in *Supplementary file 1*).

## Discussion

Proteomic experiments can be limited by sample amount or instrument sensitivity. We and others have previously presented high-throughput proteomic experiments using micro-flow-rate (*Vowinckel et al., 2018*; *Bian et al., 2020*; *Bruderer et al., 2019*; *Zelezniak et al., 2018*) and analytical flow-rate chromatography (*Messner et al., 2020*; *Messner et al., 2021*). Specifically, the latter presents highly desirable properties to increase the throughput of proteomic experiments, for example, ease of use, the ability to run fast gradients, and high reproducibility. The disadvantage of higher flow rates is however the higher dilution, resulting in a requirement for higher sample amounts and, typically, a decrease in the proteomic depth. However, new acquisition techniques can mitigate these problems.

The fast acquisition speed required to conduct proteomic experiments with fast chromatography is often reached using TOF mass spectrometers. Nonetheless, in TOF instruments, all fragment ions are recorded in parallel (quasi-simultaneously), where the ions spread by m/z across the distance between the exit of the collision cell and the TOF accelerator. When the subsequent pulse is observed at the accelerator, only a small proportion of the total fragment ion population is situated at the right location to enter the TOF, leading to a proportion of ions not being able to be detected, thus limiting sensitivity. This disproportionately impacts low-m/z ions, with the duty cycle of TOFs falling in the range of only 5–25%, depending on concentration range and m/z. Because of this duty cycle deficiency, many ions (especially low-m/z ions) are not detected in low-abundance samples, resulting in TOF-MS/MS not fully realising its full sensitivity potential. A linear ion trap introduced after the collision cell (Q2) – the Zeno trap – can compensate for these limitations in TOF-MS/MS (*Loboda and Chernushevich, 2009*). When the Zeno trap is enabled, the fragment ions in the selected mass range are trapped in an axial pseudopotential well created by an additional RF ('AC') voltage applied with the same amplitude and phase to all four rods of the trap in a required focal point. Subsequently, the release of the ions occurs by potential energy with timing aligned to the next pulse at the accelerator, enabling a >90% duty cycle, thus increasing intensity 4–20 times (*Baba et al., 2021*) without loss of mass accuracy and resolution (*Figure 1—figure supplement 1*).

To use the Zeno trap technology in high-throughput proteomics, we have established and benchmarked a DIA method termed Zeno SWATH MS. To achieve this, we modified the instrument acquisition software on a Zeno TOF 7600 system that is equipped with the Zeno trap to be able to acquire data in DIA mode, coupled the system to different chromatographic systems, and optimised acquisition parameters. We then tested Zeno SWATH MS on tryptic digests generated of protein extracts derived from a human cell line standard (K562), human plasma, as well as from yeast cultures, *E. coli* cultures, and chickpea seedling samples. Furthermore, we benchmarked the method on two-species mixtures of yeast and human cell line standard samples. In all cases, we identified more precursors and proteins, and quantified those with higher precision using Zeno SWATH MS in comparison to SWATH MS on the same instrument, as well as its predecessor, the TripleTOF 6600. We show that gains mostly result from the better identification and quantification of low-abundant precursors. Moreover, we report that Zeno SWATH MS can generate deep proteomes from low sample amounts, even when used in conjunction with 800 µl/min analytical flow-rate chromatography. For instance, on most matrices, we quantified a similar number of precursors from 62.5 ng of tryptic digest using Zeno SWATH MS as from 500 ng of the same tryptic digest using SWATH MS. Lowering the sample dependency in high-throughput proteomics has at least three different applications: First, high-throughput proteomics can be extended to studies where only low amounts of material are available. Second, injecting lower amounts of samples decreases the cleaning cycles of mass spectrometers and increases their uptime. Third, injecting lower sample amounts allows one to reduce costs. Reassuringly, Zeno SWATH MS improves both the number of proteins identified and quantification precision in all chromatographic methods tested. To this end, further applications of Zeno SWATH MS remain to be explored. For instance, our benchmark experiments were acquired with a ZenoTOF 7600 system running SCIEX OS 2.1.6, with an ionisation threshold for Zeno SWATH MS to prevent detector damage ('ion striking') caused by excessive ions hitting the detector at the same time. This detector protection procedure limits the maximum injection amount in our benchmark to 62.5 ng, preventing us from testing the effect of high amounts of sample on the MS system. The ionization threshold (which limited the load

to 62.5 ng) was experimental in nature at the time performing experiments, however, further re-optimizations to this threshold occurred enabling higher loads with Zeno SWATH MS with newer releases of SCIEX OS.

Beyond the here investigated application of Zeno SWATH MS for medium-to-high flow-rate chromatography or high-throughput proteomics, the gain in sensitivity and expansion of proteomics coverage can be useful for a range of other applications. For instance, when coupled with nanoflow chromatography systems, the sensitive Zeno SWATH MS might benefit the field of single-cell proteomics. Moreover, further improvements in sensitivity in high-throughput proteomics are still possible in the future. For instance, our lab has previously introduced a scanning SWATH acquisition technique that significantly increased the identification rate of proteins using the TripleTOF 6600 system (SCIEX), particularly in ultra-fast proteomic experiments (*Messner et al., 2021*). Scanning SWATH combines the benefits of data-dependent and data-independent techniques by employing a 'sliding' quadrupole (Q1) to assign precursor masses to MS/MS traces, allowing for a faster Q1 scan than stepwise SWATH acquisition. A combination of the Zeno trapping technology with scanning SWATH is technically possible and might result in further gains in the sensitive analysis of proteomes at high throughput.

## Materials and methods
### Materials and reagents
Water was from Merck (LiChrosolv LC-MS grade; Cat# 115333), acetonitrile was from Biosolve (LC-MS grade; Cat# 012078), 1,4-dithiothreitol (DTT; Cat# 6908.2) from Carl Roth, iodoacetamide (IAA; Bioultra; Cat# I1149) and urea (puriss. P.a., reag. Ph. Eur.; Cat# 33247) were from Sigma-Aldrich, ABC (Eluent additive for LC-MS; Cat# 40867), formic acid (LC-MS Grade; Eluent additive for LC-MS; Cat# 85178) was from Thermo Fisher Scientific, trypsin (sequence grade; Cat# V511X), K562 human cell line standard (K562, Mass Spec-Compatible Human Protein Extracts, Cat# V6951), this commercially available cell line is *Homo sapiens* origin, the identity has been authenticated, and tested by Promega, RRID: CVCL_0004. Digested yeast protein extract (Mass Spec-Compatible Yeast Protein Extracts, Cat# V7461) were from Promega, commercial human plasma samples (Human Source Plasma, LOT# 20CILP1034) were from ZenBio.

### Sample preparation
K562 human cell line standard (Promega, #V6951) and yeast tryptic digest (Promega, #V7461) stock samples were prepared via reconstitution of the powder form of the product using 200 µl of reconstitution buffer (1% ACN, 0.1% formic acid in LC grade water) and aliquoted into 4×50 µl stock samples, stored at –80°C and defrosted before sample preparation.

Plasma stock samples were prepared as follows: 5 µl of commercially available plasma sample was added to 55 µl of denaturation buffer (50 µl of 8 M urea, 100 mM ammonium bicarbonate [ABC], 5 µl of 50 mM DTT). The samples were incubated for 1 hr at room temperature before the addition of 5 µl of 100 mM IAA. After a 30 min incubation at room temperature in the dark, the samples were diluted with 340 µl of 100 mM ABC and digested overnight with 22.5 µl of 0.1 µg/µl trypsin at 37°C. The digestion was quenched by adding 50 µl of 10% formic acid. The resulting tryptic peptides were purified on a 96-well C18-based solid-phase extraction (SPE) plate (BioPureSPE Macro 96-well, 100 mg PROTO C18, The Nest Group). The purified samples were resuspended in 120 µl of 0.1% formic acid.

*E. coli* samples were prepared from $10^{10}$ cells, thawed in 200 µl lysis buffer (7 M urea, 0.1 M ABC, 2 mM $MgCl_2$, 0.125 U benzonase). The cell pellets were disrupted using a GenoGrinder in the presence of 100 mg of 0.1 mm ceramic beads. The lysate was incubated at 37°C for 15 min to ensure optimal benzonase activity. The supernatant was clarified by centrifugation at 4°C, 20,000 × *g* for 15 min.

Following addition of DTT to a final concentration of 5 mM, the samples were incubated for 90 min at room temperature before the addition of IAA to a final concentration of 10 mM. After 30 min incubation at room temperature in the dark, the samples were diluted fivefold with 100 mM ABC and digested overnight with 0.1 µg/µl trypsin at 37°C. The digestion was quenched by adding formic acid to a final concentration of 1%. The resulting tryptic peptides were purified using C18-based

(AttractSPE, Affinisep) stage tips as described by *Rappsilber et al., 2007*, and dried before resuspension in 0.1% formic acid.

Chickpea seedling sample preparation was performed essentially as described by *Wang et al., 2006*. Briefly, 500 mg of chickpea seedlings, created by the germination of chickpeas in the lab, were pulverised in liquid nitrogen. The powder was washed with trichloroacetic acid/acetone, methanol, and acetone at 4°C. After removal of acetone, the pellet was dried at 50°C for 20 min, the protein was subjected to phenol extraction and precipitated by the addition of ice-cold 0.1 M ammonium acetate in 80% methanol. The resulting pellet was washed once with ice-cold methanol and once with ice-cold 80% acetone, then air-dried. The protein pellet was dissolved in 8 M urea, 100 mM Tris, pH 8.0. Following dilution to 2 M urea, the protein was reduced with 2 mM DTT for 30 min at room temperature and alkylated for 30 min with 15 mM chloroacetamide. The sample was then digested overnight with LysC and trypsin. The resulting peptides were filtered through a 30 kDa MWCO centrifugal filter and further purified by C18 stage tips as described above.

### LFQbench hybrid samples

To generate the hybridisation of different species samples, tryptic peptides were combined as previously published (*Navarro et al., 2016*; *Meier et al., 2020*), featuring similar benchmarks justified to the maximum injectable amount of MS system to avoid ion striking, the sample mixtures were in the following ratios: MIX A was composed of 30 ng/μl of K562 human cell line standard and 35 ng/μl yeast tryptic digest; MIX B was composed of 30 ng/μl of K562 human cell line standard and 17.5 ng/μl yeast tryptic digests.

## Instrumentation

All ZenoTOF 7600 system-related methods were developed on research instruments in which an ionisation threshold for Zeno SWATH MS was introduced to prevent detector damage caused by too many ions hitting the detector at the same time ('ion striking'). SWATH window size and accumulation time were optimised and chosen to avoid ion striking while maintaining adequate precursor and protein identification numbers. The ionisation threshold has re-optimised and allow higher loads with Zeno SWATH MS with a new release of SCIEX OS.

To eliminate the potential for unreliable results due to low sample amount, before all the dilution series is acquired (as described below), we first injected replicates of blanks (50/50 v/v ACN/H$_2$O) to exclude the identification numbers that are explained by carryover, then started with the highest dilution (lowest concentration) and performed triplicate injections to prove similar results among replicates.

## ZenoTOF 7600

### Micro-flow-rate system

K562 human cell line standard dilution series and all sample species were acquired on an ACQUITY UPLC M-Class system (Waters) coupled to a ZenoTOF 7600 mass spectrometer with an Optiflow source (SCIEX). Prior to MS analysis, samples were chromatographically separated with a 20 min gradient (time, % of mobile phase B: 0 min, 3%; 0.86 min, 7.1%; 2.42 min, 11.2%; 5.53 min 15.3%; 9.38 min, 19.4%; 13.02 min, 23.6%; 15.48 min, 27.7%;17.27 min, 31.8%; 19 min, 40%; 20 min, 80% followed by re-equilibration for 10 min before the next injection; *Supplementary file 2*) on HSS T3 column (300 μm×150 mm, 1.8 μm, Waters) heated to 35°C, using a flow rate of 5 μl/min (*Zelezniak et al., 2018*) where mobile phases A and B are 0.1% formic acid in water and 0.1% formic acid in ACN, respectively. To compensate for injection volume variance, 1 μl of each dilution series sample or mixture sample was loaded prior to samples entering the MS. A SWATH acquisition scheme with 60 variable-size windows covering a precursor mass range of 400–900 m/z (*Supplementary file 2*) and 11 ms accumulation time was used. A Zeno SWATH acquisition scheme with 85 variable-size windows and 11 ms accumulation time was used (*Supplementary file 2*). For both methods, ion source gas 1 and 2 were set to 12 and 60 psi, respectively; curtain gas to 25, CAD gas to 7, and source temperature to 150°C; spray voltage was set to 4500 V. (Unique precursor and protein identification via different windows in 100 ng K562 human cell line standard injection with various SWATH acquisition schemes are shown in *Figure 5*, showing very little difference in identification performance with window number of 60 or 85.)

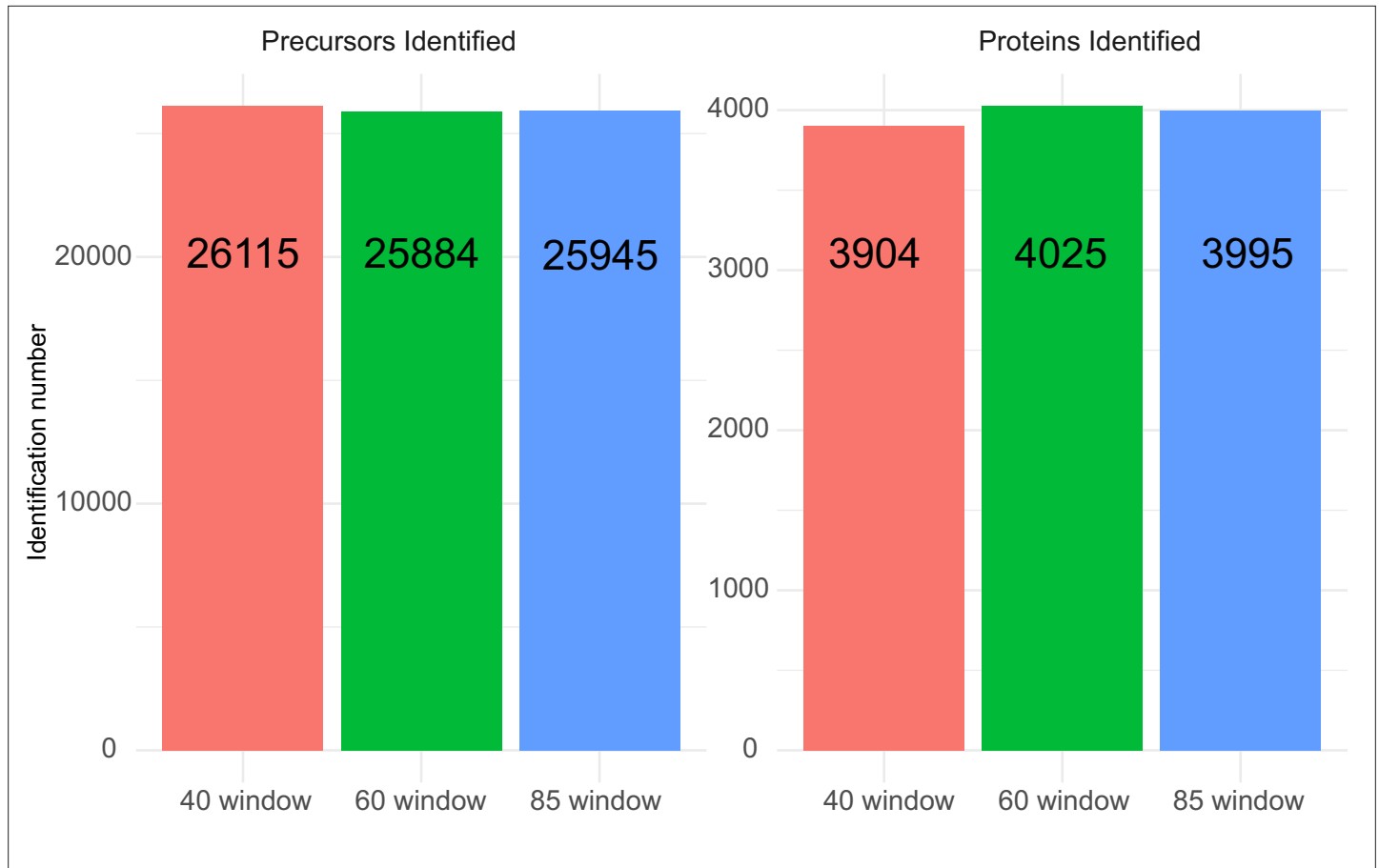

**Figure 5.** Precursor and protein identification performance with 100 ng K562 human cell line lysate injection using 5 µl/min, 20 min micro-flow chromatography coupled with different SWATH window number acquisition schemes. For the 40 variable window SWATH acquisition scheme, 26,115 precursors from 3904 proteins were identified; the 60 variable window SWATH acquisition scheme identified 25,884 precursors from 4025 proteins; and the 85 variable window SWATH acquisition scheme identified 25,945 precursors from 3995 proteins.

### Analytical flow-rate system

Standards and samples were acquired on an 1290 Infinity II UHPLC system (Agilent) coupled to a ZenoTOF 7600 mass spectrometer with a DuoSpray TurboV source (SCIEX). Prior to MS analysis, samples were chromatographically separated on an Agilent InfinityLab Poroshell 120 EC-C18 1.9 µm, 2.1 mm × 50 mm column heated to 30°C. A 5 min 0.8 ml/min flow rate of linear-gradient ramping from 3% ACN/0.1% formic acid to 40% ACN/0.1% formic acid was applied (*Messner et al., 2021*). 1 µl of each dilution series sample was loaded prior to samples entering MS. A SWATH acquisition scheme with 60 variable-size windows and 11 ms accumulation time was used (*Supplementary file 2*). The Zeno SWATH MS has the same 60 windows as the SWATH MS and 13 ms accumulation time. For both acquisition methods, ion source gas 1 (nebuliser gas), ion source gas 2 (heater gas), and curtain gas were set to 60, 65, and 55 psi, respectively; CAD gas was set to 7, source temperature to 700°C, and spray voltage to 3500 V.

### TripleTOF 6600

*Micro-flow system*: For inter-platform comparison, K562 human cell line standard dilution series were also acquired on a nanoAcquity UPLC System (Waters) coupled to a SCIEX TripleTOF 6600 mass spectrometer. The peptides were separated with the same 20 min gradient on a Waters HSS T3 column (300 µm×150 mm, 1.8 µm) using a flow rate of 5 µl/min. The SWATH MS/MS acquisition scheme was used as described previously (40 variable-size windows and 35 ms accumulation time; *Zelezniak et al., 2018*).

## Spectral libraries

The library for high-flow benchmarks using the K562 human cell line standard was generated from high pH reverse-phase fractionation of commercial K562 human cell line standard and HeLa (Thermo Fisher Scientific, #88329) digests. Individual pH fractions from each cell line were analysed with a ZenoTOF 7600 in IDA/DDA mode in-line with a Waters M-Class HPLC and searched with a Protein-Pilot application within the OneOmics suite against a human Swiss-Prot canonical and isoform FASTA file (July 2020). The ProteinPilot results from each fraction were then merged using the Extractor application in the OneOmics suite to build a spectral library for further processing of DIA results.

For the analysis of plasma samples, a public spectral library (*Bruderer et al., 2019*) was used.

## Data processing and analysis

All raw data from ZenoTOF 7600 system were acquired by SCIEX OS (v2.1.6) (note that SCIEX OS 3.0, which supports Zeno SWATH MS, is the commercial software available on the market), processed with DIA-NN (v1.8 beta 20) (*Demichev et al., 2020*) using mass accuracy of 20 and 12 ppm at the MS2 and MS1 level, respectively, scan window of 7, protein inference disabled (spectral library as described above) or relaxed (library-free, with additional command `--relaxed-prot-inf`), quantification strategy of 'Robust LC (high precision)', and library generation set as 'IDs, RT&IM profiling'. All other settings were kept at default.

The high-flow K562 human cell line lysate benchmark data was analysed using the DDA-based library described above, with MBR disabled.

The plasma acquisition analysis was performed as described previously (*Messner et al., 2020*). Specifically, the 'Deep learning-based spectra, RTs and IMs prediction' option was activated in DIA-NN, to replace all spectra and retention times in the public spectral library (*Bruderer et al., 2019*) with ones predicted in silico. Further, the protein annotation in the library was replaced using the 'Reannotate' function in DIA-NN with the annotation from the Human UniProt (*The UniProt Consortium, 2016*) isoform sequence database (UP000005640, 19 October 2021). The processing was performed using the MBR mode in DIA-NN – a two-step analysis wherein a spectral library (empirical or predicted in silico) is first refined based on the DIA experiment in question, and subsequently used to reanalyse it (*Demichev et al., 2020*).

For the analysis of micro-flow K562 human cell line standard acquisition as well as samples from other species, the respective FASTA databases from UniProt (human: UP000005640 [10 March 2022] yeast: UP000002311_559292 [28 September 2021] *E. coli*: UP000000625 [19 October 2021]; chickpea: UP000087171_3827 [22 October 2021]) were used by DIA-NN in library-free mode. Specifically, following a two-step MBR approach described previously (*Demichev et al., 2020*), an in silico spectral library is first generated by DIA-NN from the FASTA file(s); this library is then refined based on the DIA dataset and subsequently used to reanalyse the dataset, to obtain the final results.

The data were filtered in the following way. First, a 1% run-specific q-value filter was automatically applied at the precursor level by DIA-NN. Further, to compare the protein identification numbers in the dilution series between different injection amounts and LC-MS modes, a 1% global protein group q-value filter (Lib.PG.Q.Value column in the report) was also applied. For the LFQbench-type two-species benchmarks for quantification accuracy, a 1% global protein group q-value filter and 1% run-specific protein group q-value were used instead (Lib.PG.Q.Value and PG.Q.Value column in the report). We note that in any experiment processed using the MBR mode in DIA-NN, 1% global precursor q-value filtering is also applied automatically (*Demichev et al., 2020*).

CV were calculated for each protein or precursor as its empirical standard deviation divided by its empirical mean and are reported in percentages. CV values were calculated for proteins or precursors identified in all three replicate measurements.

We included the DIA-NN pipeline file in PRIDE (*Perez-Riverol et al., 2022*) PXD036786, which fully specifies the DIA-NN configuration in processing of each of the datasets analysed.

## Data availability

The mass spectrometry proteomics data have been deposited to the ProteomeXchange Consortium via the PRIDE (*Perez-Riverol et al., 2022*) partner repository with the dataset identifier PXD036786.

Dataset used for data analysis and visualisation are available online, DOI:10.17632/nwjcn4fy87.1 (*Wang et al., 2022*).

## Acknowledgements

We thank the Charité Core Facility High Throughput Mass Spectrometry for technical support, and advice in experimental design, sample preparation, data analysis, and interpretation; we thank Fatma Amari, Spyros Vernardis, and Poppy Adkin for data acquisition from TripleTOF 6600 system; Lisa Juliane Kahl and Sreejith J Varma for providing *E. coli* and *C. arietinum* seedling stocks; and Hezi Tenenboim as well as our team members for proofreading and commenting on the manuscript. This work was supported by the Ministry of Education and Research (BMBF), as part of the National Research Node 'Mass Spectrometry in Systems Medicine' (MSCoreSys), under grant agreements 031L0220 (to MR) and 161L0221 (to VD). Work in the MR lab is further supported by the Francis Crick Institute, which receives its core funding from Cancer Research UK (FC001134), the UK Medical Research Council (FC001134), the Wellcome Trust (IA 200829/Z/16/Z), and the European Research Council (ERC-SyG-2020 951475). ZW is a member of the International Max Planck Research School (IMPRS) for Infectious Diseases and Immunology. ZW's PhD studies are part of the Sonderforschungs-bereich (SFB) TRR 186.

## Additional information

### Competing interests

Ihor Batruch, Anjali Chelur, Jason Causon, Jose Castro-Perez, Stephen Tate: Employee of Sciex. The other authors declare that no competing interests exist.

### Funding

| Funder | Grant reference number | Author |
|---|---|---|
| Bundesministerium für Bildung und Forschung | 031L0220 | Markus Ralser |
| Bundesministerium für Bildung und Forschung | 161L0221 | Vadim Demichev |
| Cancer Research UK | FC001134 | Markus Ralser |
| Medical Research Council | FC001134 | Markus Ralser |
| Wellcome Trust | 200829/Z/16/Z | Markus Ralser |
| European Research Council | ERC-SyG-2020 951475 | Markus Ralser |

The funders had no role in study design, data collection and interpretation, or the decision to submit the work for publication. For the purpose of Open Access, the authors have applied a CC BY public copyright license to any Author Accepted Manuscript version arising from this submission.

### Author contributions

Ziyue Wang, Conceptualization, Data curation, Investigation, Methodology; Michael Mülleder, Conceptualization, Resources, Data curation, Funding acquisition, Investigation, Methodology, Project administration; Ihor Batruch, Resources, Software, Supervision, Investigation; Anjali Chelur, Data curation, Investigation, Methodology; Kathrin Textoris-Taube, Torsten Schwecke, Investigation, Methodology; Johannes Hartl, Investigation, Methodology, Writing - review and editing; Jason Causon, Data curation, Formal analysis, Methodology, Project administration; Jose Castro-Perez, Supervision, Methodology, Project administration; Vadim Demichev, Conceptualization, Data curation, Investigation, Methodology, Writing - review and editing; Stephen Tate, Supervision, Investigation, Methodology; Markus Ralser, Conceptualization, Resources, Supervision, Methodology, Writing - original draft, Project administration, Writing - review and editing

### Author ORCIDs

Ziyue Wang http://orcid.org/0000-0002-4121-4799
Michael Mülleder http://orcid.org/0000-0001-9792-3861
Torsten Schwecke http://orcid.org/0000-0002-9439-3508

Johannes Hartl ⓘ http://orcid.org/0000-0001-8470-5355
Jason Causon ⓘ http://orcid.org/0000-0003-2760-8115
Stephen Tate ⓘ http://orcid.org/0000-0001-5508-0819
Markus Ralser ⓘ http://orcid.org/0000-0001-9535-7413

### Decision letter and Author response

Decision letter https://doi.org/10.7554/eLife.83947.sa1
Author response https://doi.org/10.7554/eLife.83947.sa2

## Additional files

### Supplementary files

• Supplementary file 1. Precursors and proteins identification number in all sample species – precursors identification.

• Supplementary file 2. LC-MS instrumentation parameters.

• MDAR checklist

### Data availability

The mass spectrometry proteomics data have been deposited to the ProteomeXchange Consortium via the PRIDE partner repository with the dataset identifier PXD036786. Dataset used for data analysis and visualisation are available online, https://doi.org/10.17632/nwjcn4fy87.1.

The following datasets were generated:

| Author(s) | Year | Dataset title | Dataset URL | Database and Identifier |
|---|---|---|---|---|
| Wang Z | 2022 | High-throughput proteomics of nanogram-scale samples with Zeno SWATH DIA | https://doi.org/10.17632/nwjcn4fy87.1 | Mendeley Data, 10.17632/nwjcn4fy87.1 |
| Wang Z | 2022 | High-throughput proteomics of nanogram-scale samples with Zeno SWATH MS | https://www.ebi.ac.uk/pride/archive/projects/PXD036786 | PRIDE, PXD036786 |

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
