## [Editor Report]

This paper presents a novel method for high-throughput proteomic data. This is an important improvement allowing a 5-8 fold increase in detection sensitivity over methods of similar throughput. A compelling set of data acquired on human and yeast whole-cell lysate illustrate this gain in performance.

---

## [Decision Letter]

[Editors' note: this paper was reviewed by Review Commons.]

---

## [Author Response]

We thank the reviewers for their constructive feedback. We have addressed their comments and conducted additional experiments and analyses. We believe their input has helped us to greatly improve our manuscript, and we hope all edits are satisfactory.

Reviewer #1 (Evidence, reproducibility and clarity (Required)):

In this manuscript Wang et al., benchmark the new ZenoTOF with analytical and micro-flow set up and show impressive numbers of proteins identified and quantified.The paper is well written, and I have only a few minor comments:

We thank the reviewer for the positive and constructive feedback.

1. Figure 1. many of the panels are hard to read. Especially 1a.

We apologise, and agree in retrospect that original Figure 1 and specifically 1a were overloaded with information not necessary for the manuscript. Based on the reviewer’s suggestion, we simplified the instrument structure scheme and made it a separate Figure (Figure 1 – figure supplement 1 in the revised manuscript). We hope the edits improved the readability of the figure.

2. Figure 1d. can the human amount not be normalised to log2=0?

We modified Figure 1d (Figure 1c in the revised manuscript), normalised human to log_2_ = 0, and made it easier to read.

3. Please provide in the legend the bin size for the figure in 1e.

Thanks for pointing this out. As suggested, we included the bin size in the legend of Figure 1e (Figure 2c in the revised manuscript).

4. Page 10 top: did SWATH identify more proteins than Zeno SWATH in plasma? There is something wrong as the figure shows something else. Also: That sentence in brackets is confusing.

We apologise if our Figure was confusing. With Zeno SWATH we obtained a better identification number in both precursor and protein numbers compared to SWATH in plasma (as in the other matrices). We hope that the updated versions of the figures are now clear.

Moreover, we added a sentence to the text to explicitly spell out the numbers:

“In human neat plasma, with Zeno SWATH acquisition, we identified 3,947 precursors from 245 proteins, while with SWATH acquisition, 2,516 precursors from 156 proteins were identified with same 62.5 ng sample load”.

5. Typo: page 10: respectively.

Thanks for noticing, the typo was corrected.

6. Please add raw data to PRIDE or a similar repository.

We uploaded the data to PRIDE, and the accession number is # PXD036786. All data will be made public with the publication of the manuscript.

Reviewer #1 (Significance (Required)):This is an impressive new technology that has been benchmarked by the Ralser group. It outperforms current state-of-the-art approaches.The primary audience is the proteomics community.My expertise is proteomics and quantitative mass spectrometry. I am well qualified to review this paper.

We greatly thank the reviewer for the time they spent reviewing our article and their input.

Reviewer #2 (Evidence, reproducibility and clarity (Required)):

Summary:Wang et al. present an evaluation of a new generation of time-of-flight-based mass spectrometer that improves on the fraction of ions factually used for detection of peptide analytes, thus boosting the sensitivity of the Zenotof 7600 system when compared to the same instrument with the duty-cycle-enhancing Zenotrap module disabled and also when compared to the previous generation instrument of the same vendor in some of the comparisons.The authors position the MS acquisition technique as particularly suitable in combination with medium (micro-) and high ('analytical') flow and throughput methods where higher flow rates (vs. conventional nanoflow-LCMS) allow rapid sample turnover and high throughput, yet limit the efficiency of electrospray and ion transfer into the MS system, thus being in dire need for enhanced sensitivity of the MS system employed for detection. The competency of such an MS system for very low input materials as e.g. encountered in emerging single-cell proteomic workflows, typically employing nanoflow chromatography, was thus not part of the study.Accordingly, a medium- (micro-flow) and very high ('analytical'-flow) throughput LC method were screened on the three MS (parameter) setups using human cell lysate digests typically utilized in such technical evaluations. Well-received, the authors further extended their analysis (for the new instrument) across additional sample types of clinical and extended biological interest and spanning different levels of complexity and dynamic range of contained protein analytes.In addition, the authors also performed a controlled ratio 2-species mixture experiment which allows detailed benchmarking of proteome coverage as well as the quality of protein quantification in a known differential comparison for the medium throughput (micro-flow) method.The data quite convincingly demonstrate an increased sensitivity of the instrument based on similar identification performance in DIA bottom up proteomics from ca. 3- to 8-fold lower input peptide mass. However, I see a number of shortcomings mainly in the presentation and in part the completeness of the work, with specific comments below.

We thank the reviewer for the accurate summary of our manuscript, and we addressed his specific requirements, outlined below in a point to point reply.

We further would like to comment that we concentrated on medium-to-high flow rates and microflow chromatography, as we see the main application of the new acquisition technology in medium- and large-scale screening approaches in systems biology and biomedicine. We totally agree with the reviewer that single-cell biology is another exciting new application for proteomics too, yet in its earlier stages, and we would choose another setup (i.e. nano flowrate chromatography and not high flowrates) for this application. That said, we agree with the reviewer that, plausibly, Zeno trapping can be useful in single-cell proteomics as the sensitivity increases. We added a final paragraph to the Discussion section and highlight this potential application.

Major comments:– Are the key conclusions convincing?1. The concluded 10x sensitivity increase is overstating the observed numbers (x5-x8). In addition, the authors should at least discuss other changes than the Zeno trap incurred in the Zeno SWATH vs non-Zeno-SWATH DIA setups, particularly changes in accumulation times per m/z range, with Zeno Swath accumulating ~42 % longer per cycle spanning the same m/z range (85 vs 60 windows with 11ms per window) in the uflow method set and ~ 18 % longer in the high-flow method set (same window number but 13 ms vs. 11 ms dwell time per window). This should be discussed as one of the optimizations/factors contributing to the increased sensitivity observed in Zeno Swath measurements vs conventional SWATH.

We apologise for having used the placative 10-fold increased sensitivity from the precursor ID increases obtained with lower sample amounts. We revised the manuscript now considerably and express gains in IDs with specific numbers, rather than giving an ‘order of magnitude’ range.

Benchmarking two different proteomic methods has a typical caveat—that one may not be able to use the exact same settings on the different methods if one does not want to discriminate against one method. Expressed in other words – for our specific case – do I want to quantify the pure impact of the Zeno trap itself, or do I want to evaluate the performance of a workflow that uses the Zeno Trap, versus a workflow that does not use the Zeno Trap? Both are legitimate questions, but we believe the second is more relevant for most of the readers. In our specific case, several method parameters differ between SWATH and Zeno SWATH for the reason that due to the higher ion transmission, the detector saturates earlier in Zeno SWATH than it does in SWATH mode, which results in slightly different acquisition schemes being optimal for either SWATH or Zeno SWATH, respectively.

On our specific platform, when using the 60 window acquisition scheme in Zeno SWATH, using our chromatography, and when more than 62.5 ng of sample is injected, the number of ions that hit the detector exceeds a threshold set to protect the detector from damage. Once this threshold is exceeded, the instrument firmware stops acquiring. This is why we cannot inject very high amounts of digest on our instrument when running Zeno SWATH acquisition. To keep the ion counts below the ion striking threshold in Zeno SWATH, we used a smaller window size (resulting in lower ion transmission per unit time), and in order to compensate for that, we increased the window number to 85 at 62.5 ng load as well as all dilution series.

In order to address the reviewer’s point, we tested to quantify the impact of the different windows. We conducted an experiment using SWATH, injected 100 ng of sample, and acquired the data with both 40, 60, and 85 windows. Of note, we observed no strong effect on the number of SWATH windows, and the number of precursors or proteins identified, particularly with respect to 60 and 85 windows. We now clearly state differences in the number of SWATH windows between methods, and their effect in the manuscript (Figure 5).

Illustrated here is the comparison of different SWATH-MS acquisition schemes (SWATH window number 40, 60, and 85). With 100 ng of K562 standard injected, in the 40 window SWATH method, 26,115 unique precursors from 3,904 proteins were identified; 25,884 precursors from 4,025 proteins were identified by 60 window SWATH-MS; and 25,945 precursors from 3,995 proteins were identified by 85 window SWATH scheme.

On a similar note, it is also correct that the accumulation time is different between the acquisition schemes that we used for analytical flowrate and for micro flowrate chromatography, running with SWATH and Zeno SWATH MS. Also this is intentional, as the different chromatographic regimes create very different requirements on the acquisition schemes to perform optimally.

On that note, it was unclear to me when and where the 40 variable window SWATH method mentioned in the methods was used and where the settings can be found.

The 40 variable window SWATH method was used on the SCIEX 6600 TripleTOF system for comparison of identification numbers and protein coverage between the two instruments in Figure 2a-c. We apologise that due to the focus on Zeno SWATH, we kept the part about the 6600 too brief. We expanded the Methods section for the 6600:

“TripleTOF 6600:

For inter-platform comparison, K562 dilution series were also acquired on a nanoAcquity UPLC System (Waters) coupled to a SCIEX TripleTOF 6600 mass spectrometer. The peptides were separated with the same 20-min gradient on a Waters HSS T3 column (300 µm × 150 mm, 1.8 µm) using a flow rate of 5 µl/min. The SWATH-MS/MS acquisition scheme was used as described in previous work (40 variable size windows and 35 ms accumulation time [8]).”

2. Since injected material is a critical parameter here, it would be good if it was mentioned also with the key conclusion on the increased number of confidently quantified peptides in microflow (based on the 2-species controlled quantity experiment).

We thank the reviewer for the suggestion. We have added a new 2-species deconvolution experiment that shows the performance of Zeno SWATH versus SWATH with 50% lower sample amount. (See below). Furthermore, we now explain better how the injection amounts were chosen. Based on previous works featuring similar benchmarks (Navarro et al. 2016; Meier et al. 2020), we justified the total injection amount to the maximum injectable amount of the MS system and defined the 2-species mixtures ratio for the LFQbench test.

Methods “to generate the hybridisation of different species samples, tryptic peptides were combined following previous works (Navarro et al. 2016; Meier et al. 2020) featuring similar benchmarks, justified to the maximum injectable amount of MS system to avoid ion striking, the sample mixtures were in the following ratios…”

– Regarding the confidently quantified peptides, we added LFQbench based on peptides. At both peptide and protein levels, Zeno SWATH shows a strong increase in the number of confident identification (with same injection amount, SWATH quantifies 11,227 precursors from 1,741 human proteins and 4,755 precursors from 776 yeast proteins; Zeno SWATH greatly increases the quantification number to 32,946 precursors from 3,282 human proteins and 13,983 precursors from 1,692 yeast proteins) (Figure 1—figure supplement 2b,c).

– The additional experiment where we inject 50% of the sample of 2-species sample preparations is included. We find that for both species, a similar strong increase of precise quantification number of proteins from different species is observed (SWATH method: around 1.6-fold increase with double the injection amount; Zeno SWATH: around 1.3-fold increase for both species). Also, even with double the injection amount, SWATH acquisition quantifies fewer proteins compared to Zeno SWATH (Zeno SWATH precisely quantifies (valid ratios with at least two replicates in each of A and B) 2,714 human and 1,299 yeast proteins, while with double the injection SWATH quantifies 1,741 human and 776 yeast proteins). With the same injection amount, Zeno SWATH aquantifies 3,282 human and 1,692 yeast proteins, increasing the confidence quantification number around 2-fold. In general, Zeno SWATH allows more precise quantification of mixed species compared to SWATH acquisition (Figure 1 —figure supplement 2f,g).

3. Conclusions 'increasing protein identification numbers through the use of analytical-flow-rate chromatography' does not capture the observed data; the use of analytical-flow-rate does not convey an increase in protein identification numbers but enhanced sensitivity rather enables the maintenance of high protein identification numbers / proteome coverage despite/concurrent with analytical-flow chromatography.

We apologise for poor phrasing; we never aimed to suggest that high-flow-rate liquid chromatography would give higher ID numbers compared to lower flow rates – the sentence should have said that Zeno SWATH performs better than SWATH on the fast chromatography. We corrected the statement. In addition to further avoiding ambiguity, we give the examples. “For instance, with 250 ng of K562 human cell line standard injected, around 3,000 proteins were quantified with Zeno SWATH MS while a similar protein quantification number can only be reached by SWATH MS with 2µg injection on the same instrument and chromatography”.

4. In titration curve experiments like these, probing proteome coverage from relatively small sample amounts, special care.

Although the sentence is somewhat cut off, we believe the reviewer refers to the difficulties and considerations to be taken when working with small amounts of samples. We agree. The way we do these experiments is by first injecting several blanks, to exclude numbers that are explained by carryover, and then start with the highest dilution/lowest concentrated sample. The caveat is that like in any proteomic experiment, the values obtained with the lowest sample amounts are the least reliable. In part, this is mitigated in our manuscript by obtaining relatively similar results across the replicates (K562) included in our study. Nonetheless, we outlined this potential caveat in the manuscript.

Methods: “Before all the dilution series is prepared (as described below), we first inject replicates of blanks (50/50 v/v ACN/H_2_O) to exclude the identification numbers that are explained by carryover, then start with the highest dilution (lowest concentration), and performed triplicate injections to prove similar results among replicates.”

Results: “The concentration series was acquired starting with the lowest concentrations injected after several blank measurements with a 20-min, 5 µl/min chromatographic gradient (Methods), covering sample amounts ranging from near 1 ng to 500 ng (Methods).”

5. Should the authors qualify some of their claims as preliminary or speculative, or remove them altogether?

We carefully revised the paper so that we clearly separate claims from preliminary/speculative results. All key results are backed up experimentally.

5.1 'Zeno SWATH increases protein identification in complex samples 5- to 10-fold when compared to current SWATH acquisition methods on the same instrument'.5.2 At no point this is shown, a decrease of required input amounts by 5-8-fold (increase in sensitivity) is shown by the data, not a multiplication of protein identification rates by that factor.

We apologise for the poor phrasing. We have been too hand-waving in descrbing the performance gains as ‘an order of magnitude’ ‘10 fold’ as estimated by the number of IDs gains obtained in the dilution-series experiments. We adjusted the numbers as suggested by the reviewer. We now strictly avoid hand-waving statements and give precise values instead, throughout the text.

For instance:

“Interpreting the data in a different way, Zeno SWATH MS identified a similar number of precursors and proteins from 62.5 ng of K562 human cell line standard as SWATH MS did from 500 ng of the same sample on the same instrument (SWATH identified 64,316 precursors or 5,242 proteins from 500 ng)”.

6. Would additional experiments be essential to support the claims of the paper? Request additional experiments only where necessary for the paper as it is, and do not ask authors to open new lines of experimentation.6.1 Figure 1f, Supplemental Figure 1b, Figure 3 and Supplemental Figure 3 lack data for the Zeno SWATH method's performance at higher concentration. Given the fact that there is a clear, continuous trend of significant enhancement of proteomic depth in the highest 3 concentrations sampled by the Zeno SWATH method, I lack an assessment of the upper limit of proteome coverage achievable by the new platform when input material is not limited, or at least learn why injecting more is not advisable on the ZenoTOF 7600 system. It is clear that the region of interest is the lower loads where sensitivity gains are most pronounced, but with the strong trend in IDs per ng injected in the sampled range and discrepant range sampled by the non-Zeno method I feel there is a gap in the dataset and the upper ceiling of proteome coverage could be mapped out more thoroughly (At least for human cell lysate and possibly human plasma where trends appear most (log2-)linear).6.2 Similarly, unless constrained for technical or practical reasons, I would suggest to find the ceiling for achievable proteome depth in analytical flow (4, 8 ug?)

We thank the reviewer for the comment. As described above, we have already injected up to the maximum limit. On our 7600 instrument, when running. Zeno SWATH MS, cannot analyse higher sample amounts as a detector ion-count threshold is reached (‘ion striking’). We improved the discussion of this instrument limitation in the manuscript, and apologise that this technical limit of the hardware was not sufficiently discussed in the previous manuscript version. However, high amounts can be injected on the 6600 and the 7600 instruments when both are operated with SWATH-MS. This data is included, and we now discuss the ion striking limit much more predominantly.

6.3 Are the suggested experiments realistic in terms of time and resources? It would help if you could add an estimated cost and time investment for substantial experiments.

All requests by the reviewer were totally reasonable, and we conducted all experiments that were technically feasible (i.e. all but the high-sample-amount injections using Zeno SWATH MS, where the maximum was already shown in the previous data).

6.4. All these should be re-injections of existing samples on these MS setups and a minor effort provided instrument availability (<1w) and rapid re-analysis via DIA-NN.

Please see 6.1.

7. Are the data and the methods presented in such a way that they can be reproduced?

All method details are provided.

7.1 The raw data have not been deposited to a public repository. Reproducibility of the study would benefit significantly by raw data (including search results and spectral libraries with log files of creation) upload/sharing e.g. via ProteomeXchange/PRIDE.

The raw data were uploaded to PRIDE, ID#PXD036786, and will be released with the publication of the manuscript.

7.2 If any software versions or firmwares on the hardware are required to perform the measurements on the ZenoTOF on the market today, these versions and prospective release dates should be included or the accessibility of these settings commented on.

We added the version of SCIEX OS and DIA-NN. Sciex OS is commercial, while DIA-NN is free to use.

“All raw data from ZenoTOF 7600 system were acquired by SCIEX OS (v2.1.6) (SciexOS 3.0, which supports Zeno SWATH, is now commercial software available on the market) and processed with DIA-NN (v1.8 β 20, provided in the Supplementary Materials).”

8. Are the experiments adequately replicated and statistical analysis adequate?8.1 Figures 3 and Supplemental Figure 3 need a clarification in the legend as to the nature and origin of ID numbers (mean? Number of replicates? Add error bars if possible)

We thank the reviewer for the suggestion. We modified the legends accordingly. All critical benchmarks were conducted in replicates and, if applicable, also in serial dilution series. Only the validation experiments in which we confirm the gains of Zeno SWATH MS versus SWATH-MS on additional biological matrices (i.e. plant, yeast, bacterial, plasma proteome extracts) are conducted using serial dilution series only. We have provided the ID numbers in Supplementary table 1.

We have added explanation sentence in legend: “*Shown data were the identification number of respective species and concentration, detailed information of numbers can be found in Supplementary Table 1.*”

8.2 The usage of DIA-NN for data analysis is somewhat unclear, in particular the in Methods/Spectral libraries "For the analysis of plasma samples, a project-independent public spectral library [29] was used as described previously [15]. The Human UniProt [30] isoform sequence database (UP000005640, 19 October 2021) was used to annotate the library and the processing was performed using the MBR mode in DIA-NN."

We added more detailed settings of DIA-NN and subsequent filtering steps in Methods. We also included a pipeline of DIA-NN in Supplementary Materials.

“All raw data from ZenoTOF 7600 system were acquired by SCIEX OS (v2.1.6) (note that SCIEX OS 3.0, which supports Zeno SWATH MS, is the commercial software available on the market), processed with DIA-NN (v1.8 β 20) (20) using mass accuracy of 20 and 12 ppm at the MS2 and MS1 level, respectively, scan window of 7, protein inference disabled (spectral library as described above) or relaxed (library-free, with additional command --relaxed-prot-inf), quantification strategy of "Robust LC (high precision)", and library generation set as “IDs, RT& IM profiling”. All other settings were kept default.

The high-flow K562 human cell line lysate benchmark data was analysed using the DDA-based library described above, with MBR disabled.

The plasma acquisition analysis was performed as described previously (15). Specifically, the ‘Deep learning-based spectra, RTs and IMs prediction’ option was activated in DIA-NN, to replace all spectra and retention times in the public spectral library (29) with ones predicted in silico. Further, the protein annotation in the library was replaced using the ‘Reannotate’ function in DIA-NN with the annotation from the Human UniProt (33) isoform sequence database (UP000005640, 19 October 2021). The processing was performed using the MBR mode in DIA-NN—a two-step analysis wherein a spectral library (empirical or predicted in silico) is first refined based on the DIA experiment in question, and subsequently used to reanalyse it (20).

For the analysis of micro-flow K562 human cell line lysate acquisition as well as samples from other species, the respective FASTA databases from UniProt (human: UP000005640 (10 March 2022) yeast: UP000002311_559292 (28 September 2021) *E. coli*: UP000000625 (19 October 2021); chickpea: UP000087171_3827 (22 October 2021)) were used by DIA-NN in library-free mode. Specifically, following a two-step MBR approach described previously (20), an in silico spectral library is first generated by DIA-NN from the FASTA file(s); this library is then refined based on the DIA dataset and subsequently used to reanalyse the dataset, to obtain the final results.

The data were filtered in the following way. First, a 1% run-specific q-value filter was automatically applied at the precursor level by DIA-NN. Further, to compare the protein identification numbers in the dilution series between different injection amounts and LC-MS modes, a 1% global protein group q-value filter (Lib.PG.Q.Value column in the report) was also applied. For the LFQbench-type two-species benchmarks for quantification accuracy, a 1% global protein group q-value filter and 1% run-specific protein group q-value was used instead (Lib.PG.Q.Value and PG.Q.Value column in the report). We note that in any experiment processed using the MBR mode in DIA-NN, 1% global precursor q-value filtering is also applied automatically (20).

Coefficients of variation (CV) were calculated for each protein or precursor as its empirical standard deviation divided by its empirical mean, and are reported in percentages. CV values were calculated for proteins or precursors identified in all 3 replicate measurements.

We included the DIA-NN pipeline file in PRIDE (Perez-Riverol et al., 2022) PXD036786, which fully specifies the DIA-NN configuration in processing of each of the datasets analysed.”

8.3 The authors should address in a revised version whether the identification numbers reported stem from two-pass or single-pass analysis (i.e. when the feature termed Match-between-runs) implemented since DIANNv1.8 was enabled and whether all runs, spanning different injection amounts were co-analyzed and data-re-queried for a targeted library containing precursors identified in high load samples in first pass analysis and then queried in low-load samples. In other words, are the low-load IDs independent of high load IDs? If not (i.e. the different loads were co-analyzed with MBR), what proteome coverage to the low sample loads reach bona fide, without the 'guidance' of high-load IDs?

We clarified this in the revision, thank you for the suggestion. We are using double-pass analysis: MBR was enabled for a fair comparison in the dilution series. The library-free analysis by DIA-NN is generally supposed to be used with a two-pass strategy, originally described (Demichev et al. 2020) and now termed ‘MBR’ within DIA-NN. That is, ‘bone fide’ IDs during library-free search are supposed to be only used to ‘refine’ the spectral library, which will then be used for the second pass. Otherwise, the data analysis (with DIA-NN or other software) would struggle with lower injection amounts—these are typically challenging when the search space is large due to the library-free mode. We would like to highlight that although the ‘MBR’ in DIA-NN shares the name with the respective algorithm in MaxQuant, the principle of operation here is different and simpler, i.e. in DIA-NN ‘MBR’ just refers to the two-pass analysis strategy with library refinement after the first pass (Demichev et al. 2020). This way, indeed, the numbers reported for each of the runs in the dilution series are increased through the MBR strategy, by virtue of runs being analysed together. Effectively, for the lower injection amounts, this works like analysing those with a project-specific DIA-based spectral library. In order to address the reviewer’s point, we now also added a benchmark of bona fide first-pass ID numbers (Figure 3 —figure supplement 1b), which indicates an even greater advantage for Zeno SWATH over SWATH.

8.4 Side note: Turning this around, could a high-load injection e.g. from a pool of limited-amount samples serve as a guiding element in a MBR-enabled analysis of a large cohort with limited sample amounts available per biological condition?

We agree with the reviewer's point of view, thank you for the input. Indeed, this is a main intention of MBR. For example, in the original DIA-NN release (Demichev et al. 2020), we manually performed the two-step ‘MBR’ procedure to improve identifications in mixed-species runs of the LFQbench test: the injections in which yeast or *E. coli* digests were at high concentration helped to increase the IDs in the injections in which the respective digests were at low concentration. However, the concept will be expanded further and explained in a future publication that will detail the new developments inside DIA-NN: we believe a future development of the MBR feature in DIA-NN is out of the scope of this manuscript, which focuses on Zeno SWATH technology.

Minor comments:1. Specific experimental issues that are easily addressable.1.1 The authors state the impact on dynamic range of identification when comparing ID sets against an external dataset with presumable cellular concentration numbers. I would in addition suggest comparing the dynamic range of the quantitative values observed from the available data which should provide a direct assessment of the dynamic range of quantification of the two methods.

We thank the reviewer for the suggestion. We created a correlation plot for directly comparing SWATH and Zeno SWATH. Overall, the shared proteins quantified in both methods are highly correlated, with an r² of 0.93. The plot also reflects that the quantities agree extremely well correlated in the upper four orders of magnitudes of the concentration range, and that the noise largely comes from the one order of magnitude that is close to the LOD (in SWATH), were Zeno SWATH performs better. Furthermore, the correlation analysis confirms that most proteins quantified by Zeno SWATH and not SWATH are in the low abundant range (below 10E3). (Figure 2d)

2. Are prior studies referenced appropriately?2.1 The statement that conventional DIA methods rely on nanoflow chromatography (p3, paragraph 3) is not accurate as there is previous implementations of data-independent acquisition MS of microflow separations, in part the group's work and referred to later in the text.

We apologise if our text was misleading; indeed it was, to our knowledge, our own lab that published the first coupling of microLC and SWATH-MS, in a preprint in 2016 (https://doi.org/10.1101/073478), eventually published in 2018 (Vowinckel et al. 2018). Another important study was by Bian et al. 2020 (reference 16) in this section, which shows micro-flow can overcome nanoflow sample limitation with moderate loss of sensitivity.

Another important study we quote is by Bruderer et al., Analysis of 1508 plasma samples by capillary-flow data-independent acquisition profiles proteomics of weight loss and maintenance. Mol. Cell Proteom. 18, 1242-1254 (2019), which, next to our own manuscript by Zelezniak et al., Cell Systems, 2018, are among the first two studies that used a combination of microLC and DIA for processing large proteomic sample numbers. These references have been included in the manuscript.

2.2 P.3 paragraph 3 'Moreover, the increased sensitivity of DIA methods has facilitated applications in large-scale proteomics, including system-biology studies in various model organisms, disease states, and species [5-9]' Include Ref 4 where improved sensitivity of DIA was demonstrated (at proteomic breadth.)

We thank the reviewer for the suggestion, and included the reference.

3. Are the text and figures clear and accurate?3.1 Text and Figures need to be edited for typos, language, and clarity/accuracy.

We carefully checked and amended all figures and the text as suggested.

1) Abstract 'Zeno SWATH increases protein identification in complex samples5- to 10-fold when compared to current SWATH acquisition methods on the same instrument' – At no point this is shown, a drop of required input amounts by 5-8-fold (increase in sensitivity) is shown by the data, not a multiplication of protein identification rates.

We apologise for the placative language and poor phrasing; obviously the increase in ID numbers can not be used as numeric value to describe a fold-gain in sensitivity. Please see the point major comment 5.2 above. We now use the precise numbers throughout the text.

2) P. 4 paragraph 3: Use terms 'consensus' or 'shared' identifications or similar to refer to the proteins identified in all 3 replicates, rather than 'reproducible' when discussing the reproducibility of peptide and protein quantification (as contrast to reproducibility of identification).

We thank the reviewer for the comment. We rephrased accordingly.

3) P.3 paragraph 2 'selects and fragments multiple charge ions' -> multiply charged (?)

We rephrased the sentence to make it more clear.

“In bottom-up proteomics, data-dependent acquisition (DDA) selects precursors from full-scan MS1 mass spectrum for fragmentation, generating tandem (MS/MS) mass spectra that can be matched to spectra in a database.”4) P. 4 p. 1 'leading to under-detection' please clarify (leading to partial ion usage and limited sensitivity?)

We changed the wording to make it more clear.

“leading to a proportion of ions not being able to be detected, resulting in low sensitivity.”

5) P. 6 paragraph 3 'The gain in identification number of Zeno SWATH versus SWATH is mostly explained by an increased dynamic range: i.e. more low-abundance proteins are detected' – Reformulate/clarify: Is increased dynamic range of identifications against external quantities an explanation or perhaps simply the increased sensitivity with improved duty cycle?

We added a correlation plot (Figure 4) and rephrased this section to make it easier understood.

“improved duty cycle with Zeno SWATH increases intensity at MS/MS level and enhances the identification rate and quantification precision of lower abundance proteins (protein quantity in reference less than 10E3), thereby enlarging/widening the concentration range of analyte identification.”

*6) Term 'active gradient' unclear. An inactive gradient is isocratic flow. Omit 'active'. Isocratic/other portions are overhead*.

We rephrased the respective term, and instead of active gradient used *gradient time* versus ‘total runtime’, which includes washing and equilibration steps.

7) Figure 1 panel a) iteration scheme a-d) is redundant with the rest of the figure; use alternative iter scheme within panel a). Panel a) is further contains illegibly small fonts and should be edited for legibility

We thank the reviewer for the suggestion. We separated Figure 1a into a larger figure (Supplementary Figure 1 in the revised manuscript) for better visualisation and more clear explanation.

8) Revisit y-axis labels. Example: Figure 1f) 'Precursors Identificaiton' -> Precursors identified/Precursor identifications. Correct throughout manuscript.

We thank the reviewer, and have corrected the Figures accordingly.

9) ID bar graphs in all Figures: Cumulative IDs shade of grey is not properly visible, suggest alternate color scheme or add a black color outline to the bars.

We agree, and have added a black outline to make it more visible.

10) Figure 1 e) legend 'along gradient length' -> gradient time / retention time.

We changed the legend (current figure 2b) accordingly.

11) Figure 1 d) too small, trend lines mentioned in text invisible in graph. Boxplots very small.

We changed our figures accordingly.

12) There is three different terms used for the high throughput method (analytical-flow, high-flow, and another one. Please align where possible for clarity (i.e. choose 2 names for the 2 methods throughout the manuscript).

We thank the reviewer for the suggestion, and we unified the term to analytical-flow-rate.

4. Do you have suggestions that would help the authors improve the presentation of their data and conclusions?4.1 They authors may consider adding a short explanation of the term 'dynamic range coverage of identification' to contrast this from a direct assessment of dynamic range of quantitative values observed in this study.

We agree, and we apologise that apparently a confusion emerges, as both the analytical method as well as the proteome span over a broad dynamic range. We addressed this point by rewording the respective paragraphs; see minor comment 3.1 5.

4.2 2-species controlled experiment: The discrepancy of observed vs true mixing ratios suggests the data were scaled during the analysis which, with these mixture ratios, tends to distort the accuracy (i.e. generates offset of observed from true ratio. That's very likely not a pipetting error on a log scale). In other words, you may want to evaluate the raw quantitative ratios (w/o any normalization/scaling applied) which should be more reflective of true/manual pipetting ratios in light of normalization strategy incompatibily with certain species mix scenarios (compare Supplementary Figure 1 a).Note to the editor(s): This will not affect the clear benefit of Zenotrap usage demonstrated by the 2-species benchmark as is but can be considered a minor yet recommended improvement (thus here).

We thank the reviewer for the suggestion. Indeed the data normalisation might cause some bias. We also plotted the non-normalised value for LFQbench. Please find the non-normalised values below and also in (Figure 1 —figure supplement 2d,e).

Compared to the LFQ data presented in the manuscript (Figure 1c Zeno SWATH, Supplementary Figure 2a SWATH; using PG.MaxLFQ for calculation, median of human normalised to log_2_ = 0), the non-normalised LFQbench shown below did not have an obvious worse quantification in both species. Moreover, the confidently quantified proteins in both species (valid ratios with at least two replicates in each of A and B) are exactly the same with or without normalisation.

4.3 The 2-species controlled experiment can reveal more information than currently extracted and I would recommend to show Zeno Swath and Swath xy scatters, including count-scaled density distributions of the observed ratios, side-by side. This would give deeper understanding of the large impact of the Zeno SWATH method. Also, I believe I haven't seen any instrument to date delivering precise quantification over as broad a dynamic range as surmisable from Figure 1d which might be worth wile highlighting.

We thank the reviewer for the suggestion. Indeed the scatter plot turned out to be an excellent visualisation of the increased performance of Zeno SWATH versus SWATH. We created a correlation plot of the protein quantity (PG.MaxLFQ) SWATH and Zeno SWATH data in 2 species (Figure 1d–f). Overall, the proteins quantified in both methods are highly correlated, with an r² of 0.94 (human) and 0.95 (yeast). The plot also reflects that the quantities agree extremely well in the upper three orders of magnitudes of the concentration range, and only become noisier in the one order of magnitude that is close to the LOD in SWATH. We also followed the reviewer’s suggestion and created a scatter plot comparing the ratio between the protein quantity (PG.MaxLFQ) of mixture A & B. In general, Zeno SWATH shows better separation and less interference between two species.

Reviewer #2 (Significance (Required)):Wang et al. describe a technical advance in ion usage and sensitivity based on an ion-trap device storing and focusing ions for TOF-based bottom-proteomics measurements. The study demonstrates improved sensitivity relative to previous generation instrumentation and also explores the impact of the specific trap device relative to the general improvements of the remaining MS system. The work outlines a route towards high coverage proteomics at very high throughput and robustness, as desirable in clinical proteomics and prospective personalized medicine approaches. While not all sample types of interest are limited to the amounts where the strongest improvements are seen in the presented data, large scale studies across expansive cohorts will likely be rendered more practical and realistic due to reduced instrument contamination at reduced loads and also further applications beyond those discussed in the manuscript will be rendered feasible on the newer generation instrument.The improved ZenoTOF system and SWATH method follows a series of innovations in the mass spectrometry instrumentation, most notably and related the drastic improvement of ion utilization by storage e.g in a trapped ion mobility device earlier in the ion stream where, beyond an accumulation-based boost of sensitivity, ion mobility as a further biophysical properties is assessed in addition to the conventional m/z, as reviewed recently (doi: 10.1016/j.mcpro.2021.100138.). While these developments culminated and have been targeting low-flow, ultra-high sensitivity applications such as single-cell proteomics, the present study takes a different angle towards higher throughput measurements from significantly larger than single cell, but also significantly lower than historically required sample amounts that were prohibitive to a range of applications that are now easier to accomplish thanks to this and related work of the authors and others.The presented research appears of broad relevance and interest to the scientific community interested in protein abundance pattern analysis, in particular in larger (clinical) cohorts. Furthermore, the performance metrics on proteomic depth from human cell lysate digests will likely allow researchers with analytical quests other than those exemplified in the manuscript to extrapolate the ZenoTOF and Zeno SWATH suitability for their respective analytical targets.Reviewer Field of expertise/background:Quantitative proteomics. DIA mass spectrometry method & algorithm development & heavy usage. Protein Biochemistry. Molecular Biology.

References

Bruderer, Roland, Jan Muntel, Sebastian Müller, Oliver M. Bernhardt, Tejas Gandhi, Ornella Cominetti, Charlotte Macron, et al. 2019. “Analysis of 1508 Plasma Samples by Capillary-Flow Data-Independent Acquisition Profiles Proteomics of Weight Loss and Maintenance.” *Molecular & Cellular Proteomics: MCP* 18 (6): 1242–54.

Demichev, Vadim, Christoph B. Messner, Spyros I. Vernardis, Kathryn S. Lilley, and Markus Ralser. 2020. “DIA-NN: Neural Networks and Interference Correction Enable Deep Proteome Coverage in High Throughput.” *Nature Methods* 17 (1): 41–44.

Messner, Christoph B., Vadim Demichev, Daniel Wendisch, Laura Michalick, Matthew White, Anja Freiwald, Kathrin Textoris-Taube, et al. 2020. “Ultra-High-Throughput Clinical Proteomics Reveals Classifiers of COVID-19 Infection.” *Cell Systems* 11 (1): 11–24.e4.

The UniProt Consortium, Alex Bateman, Maria Jesus Martin, Claire O’Donovan, Michele Magrane, Emanuele Alpi, Ricardo Antunes, et al. 2016. “UniProt: The Universal Protein Knowledgebase.” *Nucleic Acids Research* 45 (D1): D158–69.

Vowinckel, Jakob, Aleksej Zelezniak, Roland Bruderer, Michael Mülleder, Lukas Reiter, and Markus Ralser. 2018. “Cost-Effective Generation of Precise Label-Free Quantitative Proteomes in High-Throughput by microLC and Data-Independent Acquisition.” *Scientific Reports* 8 (1): 4346.